# Study on SLM Forming Process, Residual Stress and Thermal Fatigue of 24CrNiMo Alloy Steel

**DOI:** 10.3390/ma14164383

**Published:** 2021-08-05

**Authors:** Yongsheng Zhao, Chenggang Ding, Hui Chen, Yong Chen

**Affiliations:** 1School of Materials Science and Engineering, Dalian Jiaotong University, Dalian 116028, China; 2School of Materials Science and Engineering, Southwest Jiaotong University, Chengdu 610036, China; xnrpt@swjtu.edu.cn (H.C.); yognchen@my.swjtu.edu.cn (Y.C.)

**Keywords:** 24CrNiMo alloy steel, SLM, residual stress, thermal fatigue, SR

## Abstract

The selective laser melting (SLM) forming process of 24CrNiMo alloy steel was optimized by orthogonal experiment. The density and microstructure of the sample were analyzed, and the optimized process parameters were as follows: laser power 300 W, scanning speed 530 mm/s. The 24CrNiMo alloy steel samples were prepared with optimized parameters. The relationship between residual stress and thermal fatigue and the effect of stress-relieving annealing (SR) on residual stress were analyzed. The density of the sample was found to increase at first and then to decrease with the increase of laser power and then to decrease with the increase of scanning speed. Increasing the laser power and scanning speed widened and deepened the weld. Under the optimized process window, the formability of 24CrNiMo alloy steel samples was improved significantly. The residual stress distribution was tensile stress, which had a negative effect on the thermal fatigue properties of the sample. After SR, the residual stress changed to compressive stress, which had a positive effect on the thermal fatigue properties of the samples. Compared with the deposited state, the thermal fatigue cracks were significantly shortened after SR, which was able to further promote the improvement of thermal fatigue performance. The gradient residual stress test showed that the gradient residual stress in the edge region and the central region of the deposited sample had the same trend, and decreased gradually from the surface layer to the base layer.

## 1. Introduction

The 24CrNiMo low alloys, as a branch of Cr-Ni alloys, have been widely applied in the braking systems of the high-speed trains [1,2,3] due to their high corrosion resistance and excellent wear resistance [4] as well as their thermal fatigue resistance [5]. In addition, Mo and Ni in the alloy can improve the high temperature strength and wear performance of the alloy, and Cr contributes to improve oxidation resistance. With the increase in the speed of a high-speed train, the performance requirements of the brake disc [6] are also improved accordingly. As one of the most important core components in the high-speed train braking system, the brake disc plays a vital role in the safe operation of the train. Some studies have shown that [7] the residual stress has an important influence on the thermal fatigue performance of the parts. Therefore, the measurement of the residual stress of the brake disc is of great significance in evaluating the performance and service life of the brake disc.

Selective laser melting (SLM) technology has the characteristic of a high degree of flexibility [8,9]. Compared with traditional manufacturing methods, SLM [10,11] technology can improve the mechanical properties of brake discs, and thus its engineering application value is higher [12,13]. Some scholars [14] have studied the microstructure of 24CrNiMo alloy steel parts formed by SLM and found that the scanning rate has a certain influence on the microstructure of the samples at different positions. Some researchers [15] believe that the microstructure and mechanical properties of 24CrNiMo alloy steel will be affected by scanning spacing and scanning strategy. Hajnys et al. [16] have concluded that scanning strategy plays an important role in the influence of residual stress in SLM process. Other scholars have studied the effects of process parameters and cooling rate on residual stress [17], indicating that cooling rate and temperature gradient are related to the trend of residual stress. Some researchers have verified that the ferrite-austenitic 12CrNi2 alloy steel manufactured by laser melting deposition (LMD) can enhance its strength–ductility balance after heat treatment [18]. A study has been well verified that SR could effectively enhance the mechanical properties of the alloy [19]. An article reported the effect of rescanning strategy on residual stress and distortion of two alloys manufactured by selective laser melting. The results showed that the residual stress and distortion of all specimens subjected to rescanning strategy were more serious than those of specimens without rescanning strategy [20]. However, previous studies have mainly focused on the effects of process parameters such as laser power, scanning rate and scanning strategy on the microstructure and residual stress of 24CrNiMo alloy steel, but there are few studies on the residual stress and thermal fatigue of SR on 24CrNiMo alloy steel, especially on the gradient residual stress of parts. In this paper, by using SLM technology, the optimized parameters were selected by orthogonal test, and the 24CrNiMo alloy steel samples were prepared under the optimized parameters. The relationship between surface residual stress and thermal fatigue and the evolution law of gradient residual stress were analyzed. The law of residual stress on the surface of the sample caused by SR was studied, which provided a theoretical basis for the improvement of the comprehensive properties of the material.

## 2. Test Materials and Methods

### 2.1. Powder and Substrate Materials

The powder material in this paper was 24CrNiMo alloy steel, and the powder morphology is shown in Figure 1. The particle size was 15–53 μm. The particle size distribution and nitrogen and oxygen content are shown in Table 1 and Table 2. The substrate was made of 30CrNiMo steel with a size of 250 × 250 × 30 mm^3^. The chemical composition and mechanical properties of 30CrNiMo steel and 24CrNiMo alloy steel are shown in Table 3 and Table 4.

### 2.2. Test Methods

#### 2.2.1. Orthogonal Test Methods

According to the selection requirements of density and microstructure as optimal process parameters, two orthogonal experiments were designed. The first parameter of orthogonal experiment combination was: laser power 150~450 W, scanning speed 400~1200 mm/s. The second parameter of orthogonal experiment combination was: laser power 300~360 W, scanning speed 470~630 mm/s. After the first experiment, the second orthogonal test was carried out to find the optimal combination of process parameters. The primary and preferred test parameters are shown in Table 5 and Table 6.

#### 2.2.2. Sample Preparation and Test Methods

In this paper, the 3D printing equipment used an EP-M250 metal printer (Beijing Yijia 3D Technology Co., Ltd., Beijing, China), and the forming size could reach 262 × 262 × 350 mm^3^. The maximum power of the fiber laser was 500 W, and the wavelength was 1070 nm. The other process parameters were as follows: the scanning distance was 0.11 mm, the fixed layer thickness was 50 μm, the substrate was preheated at 50 °C, and the scanning strategy was to rotate 67°. After the sample was formed, the sample was heat treated by SR, and the samples were kept at 600 °C for 2 h and cooled in the furnace. The densification test, microstructure characterization and hardness test samples were 10 × 10 × 5 mm^3^. After ultrasonic cleaning, the sample density was measured by a DX-100E (Qunlong Instrument Co., Ltd., Xiamen, China) automatic electronic densitometer. The hardness of the sample was measured by an FM-700 (Future, Tokyo, Japan) microhardness tester, and the load was 200 g. The microstructure was observed by a LeicaDMi8A (Leica microsystems Ltd., Wetzlar, Germany) optical microscope. A self-made thermal fatigue testing machine was used in the thermal fatigue test. The schematic diagram of the thermal equipment and the size of the sample are shown in Figure 2. The surface residual stress of the sample was analyzed by μ-X360n (Pulstec Co., Ltd., Tokyo, Japan) ultrafast and high precision X-ray residual stress analyzer. The gradient residual stress testing equipment used an SCGS20 (Shandong Huayun Mechanical and Electrical Technology Co., Ltd., Jinan, China) automatic gradient stress testing system, and the test standard met the requirements of GB/T31310-2014. The thickness of the gradient residual stress test was 1 mm, and the total number of test layers was 20. The dimensions of the residual stress test specimen are shown in Figure 3.

## 3. Test Results and Analysis

### 3.1. Effect of SLM Process Parameters on Density of 24CrNiMo Alloy Steel

Table 7 shows the density test results of the first orthogonal test samples. It can be seen that the density of the three groups of samples numbered 4~7, 11~14 and 17~22 have higher densities than other samples. From Figure 4a,b, it can be seen that the density of the sample increases at first and then decreases with the increase of laser power in the research range of 150~450 W and scanning speed 400~1200 mm/s, and decreases with the increase of scanning speed, and the density is relatively high between 500~600 mm/s.

### 3.2. Effect of SLM Process Parameters on Microstructure of 24CrNiMo Alloy Steel

Figure 5 shows the microstructure of the XOZ cross section of the sample under different laser scanning speeds and the same laser power (300 W). As shown in Figure 5a, when the scanning speed was 500 mm/s, the melt channel existed in the form of droplets. As the scanning speed increased, the width and depth of the weld increased slightly, and the cladding layer tended to be flat (Figure 5d,e), but when the scanning speed reached 1100 mm/s, the molten pool became larger and the cladding layer was uneven (Figure 5f).

It can also be seen in Figure 5 that there were different degrees of holes (black spots in the figure) in SLM-formed alloy steel, and the size of the holes ranged from a few microns to dozens of microns. The large-size pores were irregular, which were mainly formed at the bottom of the overlap zone between the molten channels, and there were spherical particles in some pores, indicating that these pores were mainly unfused pores. The formation of pores was the direct reason that the density of alloy steel was affected. Figure 6 shows the microstructure of the XOZ section of the sample at different laser power and the same scanning speed (600 mm/s). The effect of laser power on the morphology of cladding layer was similar to that of scanning speed. Increasing laser power widened and deepened the weld. Under the condition of 350 W laser power (Figure 6d), the cladding layer was flatter than other samples. Similarly, varying degrees of pores (black spots in Figure 6) were also found in SLM-formed alloy steel in Figure 6. When the laser power was less than 200 W, the sample forming showed a certain fluctuation, and the independent melting channel could be observed. With the increase of laser power, the fusion between the fusion channels was better, and the surface forming gradually became smoother and smoother. At the same time, the porosity also decreased significantly at medium scanning speed, indicating that the stability of the forming process was improved.

In the first orthogonal test, the samples under different laser processes showed defects such as spheroidization, pores and cracks, as shown in Figure 7. The main reasons for the formation of cracks were analyzed: at medium scanning speed and low laser power, the amount of powder melted by laser energy was insufficient, resulting in poor stability of the molten pool, and the unstable molten pool was entrained in the surrounding metal powder. In the final solidification area, it was easy to form enriched oxides or inclusions, and finally cracks were formed under the action of stress. In the first orthogonal test, because the sample still had spherical pores and other defects, the optimization test was used to optimize the process parameters again, and the macroscopic morphology of the sample is shown in Figure 8a. Figure 8b shows the results of the sample density test. Obviously, the density of No. 9 sample was the highest, reaching 99.78%. After comprehensively considering other indicators such as microstructure and density, the optimal process parameters were determined as follows: laser power 300 W, scanning speed 530 mm/s, scanning spacing d = 0.11 mm, fixed layer thickness h = 50 μm.

Under the optimized process parameters, 24CrNiMo alloy steel samples were prepared by SLM technology. The surface morphology and low-power optical microstructure of the samples are shown in Figure 9. The tensile strength and yield strength of the deposited sample reached 1199 MPa and 1053 MPa, respectively; the elongation after fracture was 10.8%, and the hardness was 365~388 HV_0.2_. Compared with the test results of other groups, the surface of the SLM formed sample under this process parameter was smooth and well formed, the microstructure was basically free of defects, such as pores, and the density was higher under this process parameter. Therefore, the optimized process parameters obtained in the experiment were reasonable.

As shown in Figure 10a, due to the forming characteristics of SLM, the martensite structure was easily formed in the alloy steel in the molten pool area. Under the reheating action of the subsequent cladding layer, the martensite was tempered to a certain extent. Due to the technological characteristics of SLM layer-by-layer stacking, the formed martensite in the molten pool changed differently with the distance from the new molten pool during the laser scanning in the next layer. The nearest structure was remelted to become part of the new molten pool. The area where the phase transition occurred when it was not melted and heated above the austenitizing temperature was the heat-affected zone, which became the dividing line between the new molten pool and the old molten pool. The structure of the old molten pool was tempered. The closer the structure was to the new molten pool, the higher the tempering temperature and the longer the tempering time and forming of lath martensite. The molten pool area near the bottom of the new cladding layer was obviously affected by heat, being heated to a higher temperature and undergoing continuous rapid cooling and solidification, and finally granular bainite was formed. Figure 10b shows the microstructure of the SR sample, which shows that the microstructure of the sample had no obvious change (500×) compared with the deposited state after SR.

### 3.3. Test Results and Analysis of Surface Residual Stress

The test results of residual stress on the surface of samples in different states are shown in Table 8. Since the scanning method was 67° rotation scanning, the sample was formed more uniformly, so the longitudinal residual stress and the transverse residual stress of the deposited sample were not much different.

The residual stress on the surface of the deposited sample was tensile stress, with a peak value of (453 MPa)~(583 MPa), which was about 50% of the yield strength of the material. After SR, the residual stress of the sample became compressive stress, and the peak value was concentrated in (−11 MPa)~(−22 MPa). The residual stress on the deposited surface was caused by the excessive temperature gradient. During the forming of the SLM, the cladding layer experienced a sharp increase and decrease in temperature within a short period of time, resulting in the temperature near the molten pool being much higher than others area, so a large temperature gradient was formed. In the subsequent cooling and solidification stage, the larger temperature gradient restricted the shrinkage and deformation of the cladding layer in the surrounding colder area, resulting in inconsistency of volume shrinkage and expansion, mutual restriction and large residual stress. It has been reported in literature that large tensile stress will lead to premature fatigue failure of parts [21,22,23]. The SR heat treatment provided an activation energy, which helped the atoms that deviated from the equilibrium position to return to the equilibrium position, and promoted the precipitation of supersaturated solutes in the lattice, so the residual stress decreased. After SR, because the long holding time and slow cooling speed of the sample after heating, the cooling rate and time of the surface layer and the core were not much different, and the stress distribution tended to be consistent.

### 3.4. Test Results and Analysis of Gradient Residual Stress

Figure 11a, shows the results of the gradient residual stress test at the center of the deposited sample. The characteristics of the residual stress distributed along the thickness direction were as follows: the gradient residual stress had periodic fluctuations along the depth direction. The residual stress in different deposition layers was all tensile stress, and the closer to the surface it was, the greater the residual stress. Figure 11b shows the gradient stress at the edge of the deposited specimen. The edge area distribution was the same as tensile stress, and the peak stress reached 730 MPa, indicating that the tensile stress at the surface edge of the sample was relatively large. Because the edge position of the sample was affected by the protective airflow and scanning strategy during the printing process, the residual stress fluctuated greatly. It can be seen from Figure 11b that the change of residual stress in the edge region of the sample was similar to that in the central area, and it gradually decreased from the surface of the sample to the base layer. The uneven distribution of temperature caused different volume shrinkage in space, resulting in gradient residual stress. After each layer was formed, it cooled and shrank until the newly laid layer of powder was formed. At this time, the heat of the newly laid powder layer flowed into the previous forming layer, and the residual stress of the existing layer and the newly laid layer were readjusted. In addition, the stress caused by the deposition in front of the deposition layer was relaxed to meet the balance. Therefore, the gradient stress of the as-deposited sample decreased layer by layer from the surface to the substrate.

### 3.5. Thermal Fatigue Test Results and Analysis

Figure 12 shows the notch morphology of the thermal fatigue specimen. The thermal fatigue crack length of the deposited sample was 464 μm, which was higher than that of the SR specimen. Some scholars [21] report that a large residual tensile stress will lead to premature fatigue failure of parts. Compressive stresses are advantageous in terms of fatigue life because they slow down crack propagation [24,25,26]. Due to the lower plasticity and toughness of the deposited state and the larger residual stress, the thermal fatigue resistance was lower. After SR, the content of small-angle grain boundary and large-angle grain boundary increased [27], which shows a stronger ability to hinder crack growth. There were different degrees of oxidation pits and reticular cracks on the surface of deposited and SR samples, in which the surface of deposited samples were seriously oxidized and serious reticular cracks appeared, so SR specimens had better thermal fatigue properties.

## 4. Conclusions

In this paper, 24CrNiMo alloy steel was formed by SLM, the residual stress distribution of deposited and SR specimens was analyzed and the relationship between residual stress and thermal fatigue was studied. The following conclusions were obtained.

(1)Through the process optimization, the optimized parameters were obtained as follows: laser power P = 300 W, scanning speed V = 550 mm/s, scanning spacing d = 0.11 mm, fixed layer thickness h = 50 μm. Under the optimized parameters, the alloy steel samples were prepared with SLM, with a density of 99.8% and a hardness of 365–388 HV_0.2_.(2)With the increase of laser power, the density of deposited samples increased at first and then decreased, and decreased with the increase of scanning speed. At a low laser power, there were spherical particles in the overlap zone of the molten channel, then the enrichment of oxides or inclusions was easily formed during the solidification of the molten pool and cracks were finally formed under the action of stress.(3)The residual stress of the deposited sample was relatively large, which was about 50% of the yield strength of the material. The residual stress distribution of the SR sample showed that the residual stress was relatively stable and compressive, which improved the thermal fatigue performance. The gradient residual stress of the deposited sample decreased gradually from the surface layer to the base layer, which was mainly due to the stress caused by the relaxed thermal cycle deposition in front of the deposition layer. After SR, the performance and service life of the brake disc was effectively improved.

## Figures and Tables

**Figure 1 materials-14-04383-f001:**
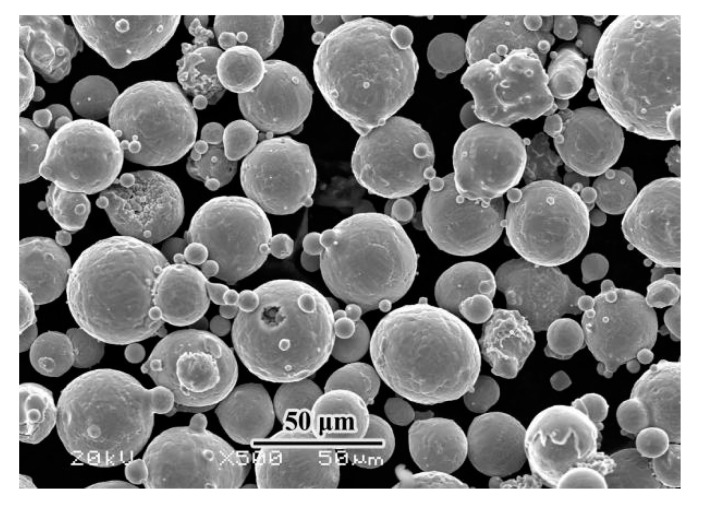
The morphology of 24CrNiMo alloy steel powder.

**Figure 2 materials-14-04383-f002:**
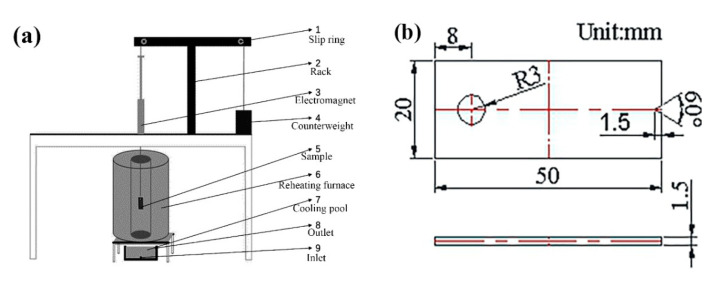
Schematic diagram of thermal fatigue equipment and sample size: (**a**) schematic diagram of thermal fatigue testing machine; (**b**) dimensions of thermal fatigue specimens.

**Figure 3 materials-14-04383-f003:**
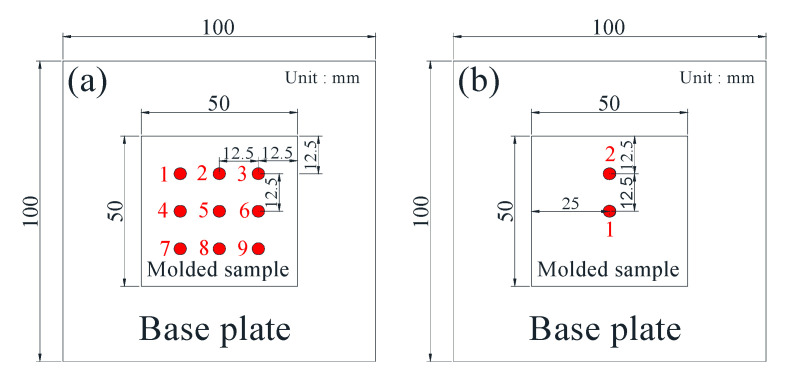
Residual stress test point: (**a**) diagram of surface residual stress test; (**b**) gradient residual stress test diagram.

**Figure 4 materials-14-04383-f004:**
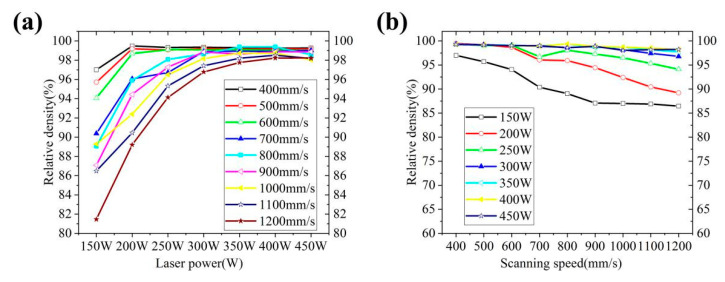
Effect of process parameters on density: (**a**) effect of laser power on density; (**b**) effect of scanning speed on density.

**Figure 5 materials-14-04383-f005:**
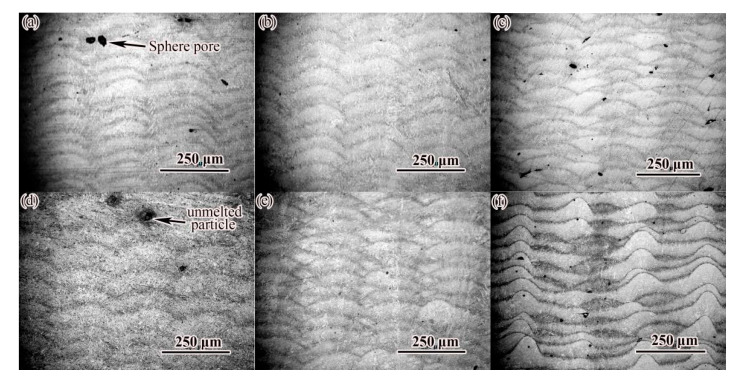
OM image of cross section of alloy steel formed by SLM at different scanning speeds: (**a**) V = 500 mm/s; (**b**) V = 600 mm/s; (**c**) V = 700 mm/s; (**d**) V = 800 mm/s; (**e**) V = 900 mm/s; (**f**) V = 1000 mm/s.

**Figure 6 materials-14-04383-f006:**
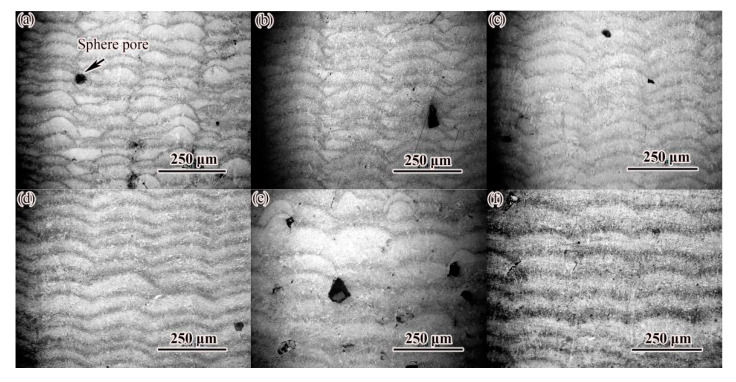
OM image of cross section of alloy steel formed by SLM at different laser power: (**a**) P = 200 W; (**b**) P = 250 W; (**c**) P = 300 W; (**d**) P = 350 W; (**e**) P = 400 W; (**f**) P = 450 W.

**Figure 7 materials-14-04383-f007:**
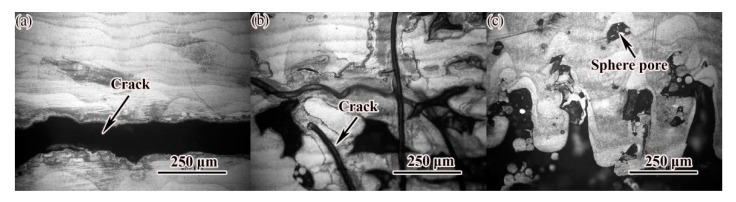
Types of defects at low laser power: (**a**) P = 150 W; V = 700 mm/s; (**b**) P = 150 W; V = 700 mm/s; (**c**) P = 150 W; V = 700 mm/s.

**Figure 8 materials-14-04383-f008:**
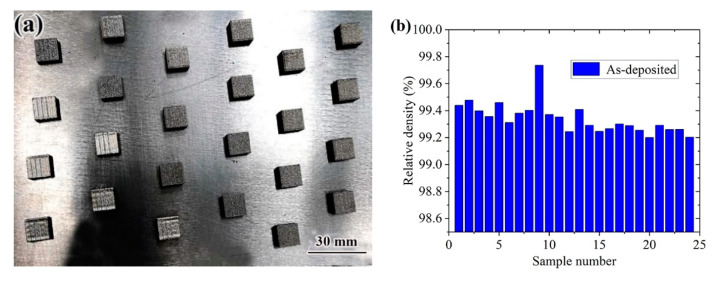
Appearance of the second batch of orthogonal specimens and densification of samples: (**a**) appearance morphology; (**b**) densification of different specimens.

**Figure 9 materials-14-04383-f009:**
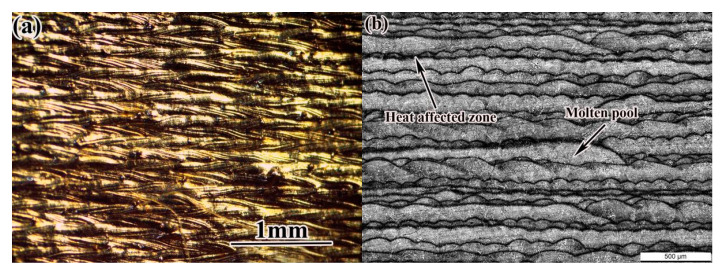
Surface morphology and microstructure of as-deposited sample: (**a**) surface morphology; (**b**) microstructure.

**Figure 10 materials-14-04383-f010:**
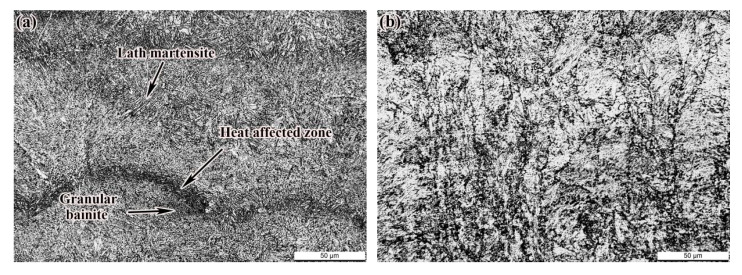
Micro-morphology of the sample: (**a**) as-deposited; (**b**) SR.

**Figure 11 materials-14-04383-f011:**
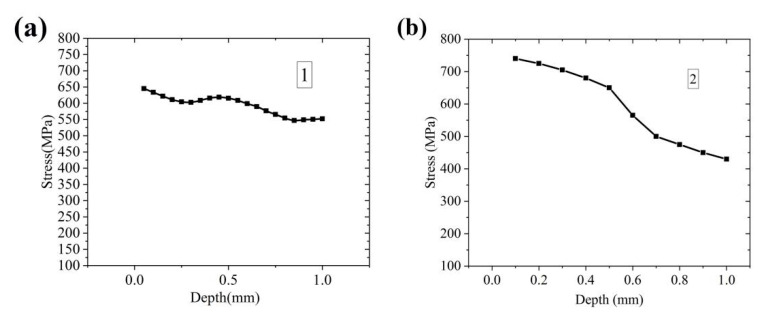
Test results of gradient residual stress: (**a**) central position; (**b**) edge position.

**Figure 12 materials-14-04383-f012:**
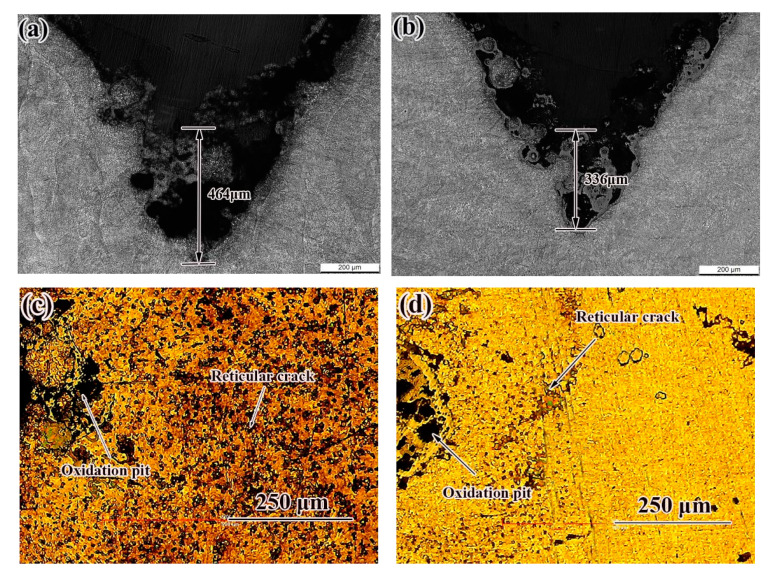
Crack morphology of samples in different states after thermal fatigue test: (**a**,**c**) as-deposited; (**b**,**d**) SR.

**Table 1 materials-14-04383-t001:** Particle size distribution of 24CrNiMo alloy steel powder.

D10/μM	D50/μM	D90/μM	Testing Reference Standard	Preparation Method
20.8	35.0	57.2	GB/T14265	Gas atomization pulverization

**Table 2 materials-14-04383-t002:** Nitrogen and oxygen content of 24CrNiMo alloy steel powder.

O/ppm	N/ppm	Testing Reference Standard
256	39	GB/T 19077

**Table 3 materials-14-04383-t003:** Chemical composition of 24CrNiMo powder and 30CrNiMo plate (%).

Grade	Fe	C	Mn	Ni	Mo	Si	Cr	S
24CrNiMo	Bal	0.23	0.72	1.81	0.47	0.21	1.12	0.0031
30CrNiMo	Bal	0.31	0.68	0.02	0.16	0.26	0.98	0.004

**Table 4 materials-14-04383-t004:** Mechanical properties of 24CrNiMo alloy steel and 30CrNiMo steel.

Grade	Yield Strength (MPa)	Tensile Strength (MPA)	Elongation after Fracture (%)	Hardness (HV10)
24CrNiMo	960	1073	11.42	371
30CrNiMo	986	1086	10.96	378

**Table 5 materials-14-04383-t005:** Parameters of the first orthogonal test.

	V(mm/s)	400	500	600	700	800	900	1000	1100	1200
P(W)										
150		1	8	15	22	29	36	43	50	57
200		2	9	16	23	30	37	44	51	58
250		3	10	17	24	31	38	45	52	59
300		4	11	18	25	32	39	46	53	60
350		5	12	19	26	33	40	47	54	61
400		6	13	20	27	34	41	48	55	62
450		7	14	21	28	35	42	49	56	63

**Table 6 materials-14-04383-t006:** Parameters of the second orthogonal test.

	V(mm/s)	470	500	530	570	600	630
P(W)							
300		1	5	9	13	17	21
320		2	6	10	14	18	22
340		3	7	11	15	19	23
360		4	8	12	16	20	24

**Table 7 materials-14-04383-t007:** Test results of density in the first orthogonal test.

N	Density (%)	N	Density (%)	N	Density (%)	N	Density (%)	N	Density (%)
1	96.99	14	99.17	27	98.90	40	98.63	53	97.42
2	99.08	15	94.05	28	99.01	41	98.87	54	98.20
3	99.02	16	98.68	29	89.05	42	98.88	55	98.51
4	99.34	17	99.09	30	95.89	43	87.00	56	98.18
5	99.30	18	99.07	31	98.08	44	92.39	57	86.45
6	99.24	19	99.09	32	98.64	45	96.49	58	89.21
7	99.26	20	99.02	33	98.37	46	98.16	59	94.12
8	95.70	21	99.03	34	98.36	47	98.73	60	96.78
9	99.18	22	90.37	35	98.58	48	98.72	61	97.75
10	99.06	23	96.06	36	87.08	49	98.04	62	98.21
11	99.21	24	96.71	37	94.44	50	86.88	63	98.23
12	99.17	25	98.90	38	97.27	51	90.45		
13	99.18	26	98.93	39	98.85	52	95.32		

**Table 8 materials-14-04383-t008:** Residual stress of different treated specimens.

Sample	Residual Stress (MPa)	Test Location								
		1	2	3	4	5	6	7	8	9
Deposited	σ_x_	495	482	562	549	524	554	539	505	500
	σ_y_	583	566	507	518	522	525	453	477	556
SR	σ_x_	−17	−5	−29	−13	−36	−17	−24	−32	−17
	σ_y_	−25	−6	−35	−2	−36	−14	−11	−11	−23

## Data Availability

Not applicable.

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
