# Peer review of "Study on SLM Forming Process, Residual Stress and Thermal Fatigue of 24CrNiMo Alloy Steel"

_materials, 2021, doi:10.3390/ma14164383_

Round 1
Reviewer 1 Report
In my opinion Fig. 7 should be described in more detail.
How was prepared the sample for TEM and what type of TEM was used?
Author Response
Word

Reviewer 2 Report
The research is very interesting but it is very poorly presented and needs extensive editing:
-clear identification of steel used (standard?)
-caption o Fig 1
-Table 2, formatting and almost all Tables
-caption of Figure 2 (identification of machine elements)
-Figure 10 and 11: one picture missing
Finally, there is not many results of thermal fatigue, therefore is the title of the paper adequate?
Author Response
Word

Reviewer 3 Report
density and residual stress stories are good
microstructure characterisation is weak, I suggest to remove TEM and spend more time on microstructure in the next paper
thermal fatigue section is also weak, so, probably, the word "fatigue" should not appear in the title
minor comments are in the file

Author Response
Word

Reviewer 4 Report
Title: SLM forming process of 24CrNiMo alloy steel and the effect of residual stress on thermal fatigue
***Global Commets
Interesting paper with appropriate methodology for research and interesting conclusions.
The introduction could be improved with a more complete literature review. Improve the reference numbers.
Conclusions can be improved.
The article needs improvement, especially in terms of formatting and editorial writing
***Commets by chapter
1.Introduction
The introduction could be improved with a more complete literature review. Improve the reference numbers.
2.Test materials and methods
Page 2- line 63.
Correct the designation of figure 1
Page 4- line 107.
Better format the legend: 1 slip ring 2 rack 3 electromagnet 4 counterweight 5 sample 6 reheating furnace 7 cooling pool 8 output 9 input
3.Test results and analysis
Page 4- line 107.
Put side by side the figures 4a) and fire 4b)
Page 8- line 176.
Gama de eixos yy Figura 8 "Densidade relativa (%)": Diminuir a gama deste eixo de [98,5% a 100,0%]
4.Conclusion
Conclusions can be improved.
References
Page 13.
Many of the references are incorrectly referenced. The name of the journal where it is published is missing

Author Response
Word
